# Acceleration of Approximate Matrix Multiplications on GPUs

**DOI:** 10.3390/e25081130

**Published:** 2023-07-27

**Authors:** Takuya Okuyama, André Röhm, Takatomo Mihana, Makoto Naruse

**Affiliations:** Department of Information Physics and Computing, Graduate School of Information Science and Technology, The University of Tokyo, Tokyo 113-8656, Japan; roehm@g.ecc.u-tokyo.ac.jp (A.R.); takatomo_mihana@ipc.i.u-tokyo.ac.jp (T.M.); makoto_naruse@ipc.i.u-tokyo.ac.jp (M.N.)

**Keywords:** approximate calculation, approximate matrix multiplication, GPU computing

## Abstract

Matrix multiplication is important in various information-processing applications, including the computation of eigenvalues and eigenvectors, and in combinatorial optimization algorithms. Therefore, reducing the computation time of matrix products is essential to speed up scientific and practical calculations. Several approaches have been proposed to speed up this process, including GPUs, fast matrix multiplication libraries, custom hardware, and efficient approximate matrix multiplication (AMM) algorithms. However, research to date has yet to focus on accelerating AMMs for general matrices on GPUs, despite the potential of GPUs to perform fast and accurate matrix product calculations. In this paper, we propose a method for improving Monte Carlo AMMs. We also give an analytical solution for the optimal values of the hyperparameters in the proposed method. The proposed method improves the approximation of the matrix product without increasing the computation time compared to the conventional AMMs. It is also designed to work well with parallel operations on GPUs and can be incorporated into various algorithms. Finally, the proposed method is applied to a power method used for eigenvalue computation. We demonstrate that, on an NVIDIA A100 GPU, the computation time can be halved compared to the conventional power method using cuBLAS.

## 1. Introduction

Computing matrix products is a fundamental and essential operation in various information-processing tasks [1,2,3]. For example, one can obtain the largest eigenvalue and eigenvector of a positive definite matrix A by repeating the following calculations:(1)$$\begin{array}{c} \left\{\begin{array}{cc} y & \leftarrow Ax\\ x & \leftarrow \frac{y}{∥y∥} \end{array}\right.. \end{array}$$
In the power method, the computation of the matrix product is responsible for most of the computational burden. The same applies to optimization algorithms [4,5,6]. Therefore, it is crucial to reduce the computation time of matrix products to speed up various scientific and practical calculations.

A well-known method for accelerating matrix multiplication is to use GPUs. Other research has focused on implementing fast matrix multiplication libraries [7,8,9], designing custom hardware to accelerate specific classes of matrix multiplication [10,11,12], accelerating distributed matrix multiplication [13,14,15], designing exact and low-computation-order algorithms based on the divide-and-conquer method [16,17,18], and designing efficient approximate matrix multiplication (AMM) algorithms [19,20,21]. These areas are still active research fields. Recently, an AI-powered method that utilizes reinforcement learning to discover matrix arithmetic algorithms that are computationally less expensive than existing methods has been reported [22], building upon a previous study by Strassen [16].

For practical purposes, in some cases, the exact result of the matrix product may be optional. In this paper, we propose a method for such a case. It is known that when only an approximate result of the matrix product is required, the AMM methods can significantly reduce the computation time compared to other matrix acceleration approaches. MADDNESS, proposed in [20], provides high-speed approximate results. The paper demonstrated that MADDNESS, when executed on a CPU, can reduce the computation time by a factor of 100 compared to the exact matrix product while retaining sufficient computational accuracy to enable neural network inference. Allowing approximate calculations enables MADDNESS to achieve an overwhelming speed-up with other matrix optimization approaches. However, GPU computation can typically perform matrix products two orders of magnitude faster than CPU computation. Therefore, MADDNESS is practical for running neural networks in environments like the Internet of Things (IoT), where power consumption and installation space limitations make GPUs infeasible. On the other hand, if GPUs are available and there are no such limitations, obtaining the exact result by GPU computation rather than an approximate result by MADDNESS on a CPU is preferable.

A significant body of research proposing custom hardware exists to improve matrix multiplication performance. Hardware research often focuses on reducing energy consumption [23,24,25,26,27,28]. Existing hardware research has introduced various methods to reduce the energy consumption of artificial intelligence (AI) algorithms while maintaining acceptable accuracy. One approach involves approximate computing, which trades computational accuracy for improved energy efficiency. This strategy is particularly effective for AI algorithms that exhibit inherent error tolerance. Several papers have proposed using approximate multipliers, simplifying multiplication operations, and reducing energy consumption. Another approach involves using logarithmic multipliers, simplifying multiplication operations while sacrificing some degree of accuracy. This research significantly advances energy-efficient computing techniques tailored to AI algorithms, a central and pertinent topic within computer science. Our proposed method pertains to the approximate matrix multiplication algorithm and does not result in a reduction in the power consumption of a single FMA operation. However, it can potentially reduce the overall computation time of the application, thus making a valuable contribution to energy conservation.

To the best of our knowledge, no prior research other than the method for sparse matrices [29] has endeavored to accelerate approximate computations on GPUs, despite their capability for fast and precise computation of matrix products. Computations that can be efficiently accelerated by GPUs typically involve a large number of parallel executions of simple operations with regular memory access patterns. MADDNESS proposes an alternative approach to matrix multiplication by using hashing trees for table look-ups, which is considered unsuitable for GPU implementation. In addition, FD-AMM [30] incorporates the frequent direction algorithm [31] into AMM. However, the previous study [30] has reported that FD-AMM requires an SVD iteration to generate a sketch matrix whenever one of the multiplication matrices changes, resulting in longer computation times than existing methods. Thus, traditional AMM algorithms, which lack design considerations for parallel processing efficiency on GPUs, cannot guarantee the expected acceleration simply by implementing them on GPUs. Nevertheless, given that GPUs are already employed to expedite matrix products in numerous applications, it is also advantageous to leverage such devices for approximate calculations. Accordingly, we identified a known Monte Carlo AMM that can exploit the parallel computing capabilities of GPUs. Additionally, we devised an algorithm to enhance the approximation performance of this extant method while minimizing the increase in computational complexity. Our numerical experiments demonstrate that the proposed approach can expedite the computation of eigenvalues, which is crucial in both scientific and engineering domains.

## 2. Proposed Methods

### 2.1. Theory of Monte Carlo AMM

In this section, we present the theory behind the Monte Carlo method [32] used to compute the approximate matrix multiplication (AMM) of matrices $A\in R^{m\times k}$ and $B\in R^{k\times n}$. We denote the *l*th column and row of a real matrix M by $M^{\left(l\right)}$ and $M_{\left(l\right)}$, respectively. The Frobenius norm of the matrix M is denoted by ∥M∥, defined as the square root of the sum of the squares of all elements in the matrix. Furthermore, the Euclidean norm of vector v is denoted as $\left|v\right|$.

Algorithm 1 presents the pseudocode for the Monte Carlo AMM. It is a randomized algorithm that returns an approximate solution Z that is close to the exact solution AB with high probability. The expected square Frobenius error between AB and Z is evaluated using the following equation:(2)$$\begin{array}{c} E\left[{∥\mathrm{AB}-Z∥}^{2}\right]=\frac{1}{c}\left\{\sum_{l=1}^{k}\frac{{\left|A^{\left(l\right)}\right|}^{2}{\left|B_{\left(l\right)}\right|}^{2}}{p_{l}}-{∥\mathrm{AB}∥}^{2}\right\}. \end{array}$$
**Algorithm 1:**  Monte Carlo AMM [32]**Require:**  $A\in R^{m\times k}$ and $B\in R^{k\times n}$, and c∈N**Ensure:** $Z\in R^{m\times n}$ such that ∥AB−Z∥ is small  1:   Z←O  2:  Set $p_{1},\ldots ,p_{k}$ such that $\sum_{l=1}^{k}p_{l}=1$  3:  **for** t=1,…,c 
**do**  4:        Pick $i_{t}$ with $P\left[i_{t}=l\right]=p_{l}$  5:        $Z\leftarrow Z+\frac{A^{\left(i_{t}\right)}B_{\left(i_{t}\right)}}{cp_{i_{t}}}$  6:  **end for**

It has been proven that this equation is minimized when the probability $p_{l}$ of sampling the *l*th column/row is set to its optimal value, defined as
(3)$$\begin{array}{c} p_{l}^{\mathrm{opt}}:=\frac{\left|A^{\left(l\right)}\right|\left|B_{\left(l\right)}\right|}{\sum_{l^{\prime }=1}^{k}\left|A^{\left(l^{\prime }\right)}\right|\left|B_{\left(l^{\prime }\right)}\right|}. \end{array}$$
In this case, the error is given by
(4)$$\begin{array}{c} {\left.E\left[{∥\mathrm{AB}-Z∥}^{2}\right]\right|}_{p_{l}=p_{l}^{\mathrm{opt}}}=\frac{1}{c}\left\{{\left(\sum_{l=1}^{k}\left|A^{\left(l\right)}\right|\left|B_{\left(l\right)}\right|\right)}^{2}-{∥\mathrm{AB}∥}^{2}\right\}. \end{array}$$
The statement that $\mathrm{AB}=\sum_{l=1}^{k}A^{\left(l\right)}B_{\left(l\right)}$ and Equation (Equation 3) imply that Algorithm 1 with the optimal Monte Carlo sampling is designed to assign a higher probability to columns/rows with larger Frobenius norms during sampling and to compute their product as an approximate solution.

### 2.2. AMM with Preprocessing: Proposed Methods

A straightforward method to reduce the expected squared error is to increase the number of sampling iterations *c*. The error decreases inversely with the number of iterations. However, increasing *c* results in a longer computation time for the Monte Carlo AMM. Therefore, in this paper, we propose a method to reduce the expected squared error without significantly increasing the computation time by further exploiting the information used in the conventional AMM.

Let $\alpha_{i}$ and $\beta_{i}$ be the mean of $A^{\left(i\right)}$ and $B_{\left(i\right)}$, respectively. We define the following two matrices
(5)$$\begin{array}{c} X:=A-e_{m}{\left(\alpha +x\right)}^{⊤},\;Y:=B-\left(\beta +y\right)e_{n}^{⊤}, \end{array}$$
where $\alpha :=[\alpha_{1},\ldots ,\alpha_{k}{]}^{⊤}$, $\beta :=[\beta_{1},\ldots ,\beta_{k}{]}^{⊤}$, $x:=[x_{1},\ldots ,x_{k}{]}^{⊤}$, $y:=[y_{1},\ldots ,y_{k}{]}^{⊤}$, and $e_{l}$ is an *l*-dimensional vector whose elements are all one. These mean that we obtain X and Y by subtracting a constant value $\alpha_{i}+x_{i}$ from the *i*th column of A and $\beta_{i}+y_{i}$ from the *i*th row of B. Then,
(6)$$\begin{array}{cc} \mathrm{AB}\; & =\left(X+e_{m}{\left(\alpha +x\right)}^{⊤}\right)\left(Y+\left(\beta +y\right)e_{n}^{⊤}\right)\\ & =\mathrm{XY}+\left(X\left(\beta +y\right)\right)e_{n}^{⊤}+e_{m}\left({\left(\alpha +x\right)}^{⊤}Y\right)+e_{m}\left({\left(\alpha +x\right)}^{⊤}\left(\beta +y\right)\right)e_{n}^{⊤}. \end{array}$$
The second, third, and fourth terms in Equation (Equation 6) can be computed using the matrix–vector and inner products instead of matrix–matrix products. By leveraging this fact, these terms can be accurately calculated with fewer computations for given matrices A, B, x, and y. Consequently, we utilize this approach to evaluate an AMM of matrices X and Y instead of directly using A and B. Then, the expected error of this approximation relies solely on the error of XY. To simplify the notation, we define $A^{\prime }:=A-e_{m}\alpha^{⊤}$ and $B^{\prime }:=B-\beta e_{n}^{⊤}$. Notably, the matrices $A^{\prime }$ and $B^{\prime }$ are constant, and the crucial observation is that all column sums of $A^{\prime }$ and row sums of $B^{\prime }$ are equal to zero.

Algorithm 2 presents the pseudocode of our proposed method. The only differences between the proposed and conventional method lie in the second and third lines. The second line solely involves matrix–vector multiplication and inner product computation, which are less computationally intensive than matrix multiplication. Furthermore, the third line allows parallel execution for each pair of *i* and *j*. Consequently, the proposed method can be regarded as integrating the less computationally demanding preprocessing employed by the existing Monte Carlo AMM. This algorithm represents the calculation of an approximate solution to AB, as shown by the following calculation.
(7)$$\begin{array}{cc} \mathrm{AB}\; & =A^{\prime }B^{\prime }+\left(A^{\prime }\beta \right)e_{n}^{⊤}+e_{m}\left(\alpha^{⊤}B^{\prime }\right)+e_{m}\left(\alpha^{⊤}\beta \right)e_{n}^{⊤}\\ \; & =A^{\prime }B^{\prime }+\left(A\beta \right)e_{n}^{⊤}+e_{m}\left(\alpha^{⊤}B\right)-e_{m}\left(\alpha^{⊤}\beta \right)e_{n}^{⊤} \end{array}$$

**Algorithm 2:** Proposed method: Monte Carlo AMM with preprocessing**Require:**  $A\in R^{m\times k}$ and $B\in R^{k\times n}$, and c∈N**Ensure:**  $Z\in R^{m\times n}$, such that ∥AB−Z∥ is small  1:  Z←O  2:  $v_{c}\leftarrow A\beta$, $v_{r}\leftarrow \alpha^{⊤}B$, $w\leftarrow \alpha^{⊤}\beta$  3:  $Z_{ij}\leftarrow Z_{ij}-w+{\left[v_{c}\right]}_{i}+{\left[v_{r}\right]}_{j}$ for all 1≤i≤m and 1≤j≤n  4:  Set $p_{1},\ldots ,p_{k}$ such that $\sum_{l=1}^{k}p_{l}=1$  5:  **for** 
t=1,…,c 
**do**  6:        Pick $i_{t}$ with $P\left[i_{t}=l\right]=p_{l}$  7:       $Z\leftarrow Z+\frac{A^{\prime (i_{t})}B_{\left(i_{t}\right)}^{\prime }}{cp_{i_{t}}}$  8:  **end for**

Equation (Equation 7) can be regarded as the case of x=y=0 in Equation (Equation 6). Remarkably, we can prove theoretically that the expected error by Algorithm 2 is equal to or less than the error by Algorithm 1. Considering the discussion on Equation (Equation 4), we see that the expected squared error for computing AB using Equation (Equation 6) and the combination of AMM with exact computations is proportional to   
(8)$$\begin{array}{cc} E\;: & ={\left(\sum_{l=1}^{k}\left|X^{\left(l\right)}\right|\left|Y_{\left(l\right)}^{⊤}\right|\right)}^{2}-{∥\mathrm{XY}∥}^{2}\\ & ={\left(\sum_{l=1}^{k}\left|A^{\prime (l)}-x_{l}e_{m}\right|\left|B_{\left(l\right)}^{\prime ⊤}-y_{l}e_{n}\right|\right)}^{2}-{∥\left(A^{\prime }-e_{m}x^{⊤}\right)\left(B^{\prime }-ye_{n}^{⊤}\right)∥}^{2}. \end{array}$$
The expected squared error can be represented by E/c. Hereafter, our objective is to establish the superior accuracy of the proposed method compared to the original Monte Carlo AMM. To achieve this, we aim to demonstrate that, through the presentation of several theorems, the global minimum of *E* occurs at x=y=0.

**Theorem** **1.**
*Equation (Equation 8) is minimized at $x_{i}=0$ if all elements in $A^{\prime (i)}$ are zero.*


**Proof.** Suppose that all elements of $A^{\prime (i)}$ are zero for a certain *i*. Let $z:=[z_{1},\ldots ,z_{k}{]}^{⊤}$ be the vector, such that
(9)$$\begin{array}{c} z_{l}=\left\{\begin{array}{cc} x_{l} & \left(l\neq i\right)\\ 0 & \left(l=i\right) \end{array}\right.. \end{array}$$
For simplicity, we introduce the following notations.
(10)$$\begin{array}{cccc} f_{1}\left(x,y\right)\; & : & = & {\left(\sum_{l=1}^{k}\left|A^{\prime (l)}-x_{l}e_{m}\right|\left|B_{\left(l\right)}^{\prime ⊤}-y_{l}e_{n}\right|\right)}^{2} \end{array}$$
(11)$$\begin{array}{cccc} f_{2}\left(x,y\right)\; & : & = & -{∥\left(A^{\prime }-e_{m}x^{⊤}\right)\left(B^{\prime }-ye_{n}^{⊤}\right)∥}^{2} \end{array}$$
Since we have $\left|A^{\prime (i)}-z_{i}e_{m}\right|=0$, the following equalities hold.
(12)$$\begin{array}{cc} f_{1}\left(x,y\right)-f_{1}\left(z,y\right)\; & ={\left(\sum_{l}\left|X^{\left(l\right)}\right|\left|Y_{\left(l\right)}^{⊤}\right|\right)}^{2}-{\left(\sum_{l\neq i}\left|X^{\left(l\right)}\right|\left|Y_{\left(l\right)}^{⊤}\right|\right)}^{2}\\ \; & ={\left|X^{\left(i\right)}\right|}^{2}{\left|Y_{\left(i\right)}^{⊤}\right|}^{2}+2\left|X^{\left(i\right)}\right|\left|Y_{\left(i\right)}^{⊤}\right|\sum_{l\neq i}\left|X^{\left(l\right)}\right|\left|Y_{\left(l\right)}^{⊤}\right| \end{array}$$
In addition, we have
(13)$$\begin{array}{cc} f_{2}\left(x,y\right)-f_{2}\left(z,y\right)= & {∥\sum_{l\neq i}X^{\left(l\right)}Y_{\left(l\right)}∥}^{2}-{∥\sum_{l}X^{\left(l\right)}Y_{\left(l\right)}∥}^{2}\\ = & {∥\sum_{l\neq i}X^{\left(l\right)}Y_{\left(l\right)}∥}^{2}-\left({∥\sum_{l\neq i}X^{\left(l\right)}Y_{\left(l\right)}∥}^{2}+{∥X^{\left(i\right)}Y_{\left(i\right)}∥}^{2}+2\langle \sum_{l\neq i}X^{\left(l\right)}Y_{\left(l\right)},X^{\left(i\right)}Y_{\left(i\right)}\rangle \right)\\ = & -{∥X^{\left(i\right)}Y_{\left(i\right)}∥}^{2}-2\langle \sum_{l\neq i}X^{\left(l\right)}Y_{\left(l\right)},X^{\left(i\right)}Y_{\left(i\right)}\rangle , \end{array}$$
where 〈·,·〉 denotes the Frobenius inner product.We can easily see that ${\left|v\right|}^{2}{\left|w\right|}^{2}={∥vw^{⊤}∥}^{2}$ holds for vectors v and w. Combining this equality, the Cauchy–Schwarz inequality, and the triangle inequality, we obtain
(14)$$\begin{array}{cc} \langle \sum_{l\neq i}X^{\left(l\right)}Y_{\left(l\right)},X^{\left(i\right)}Y_{\left(i\right)}\rangle \; & \leq ∥\sum_{l\neq i}X^{\left(l\right)}Y_{\left(l\right)}∥∥X^{\left(i\right)}Y_{\left(i\right)}∥\\ \; & =\left|X^{\left(i\right)}\right|\left|Y_{\left(i\right)}\right|∥\sum_{l\neq i}X^{\left(l\right)}Y_{\left(l\right)}∥\leq \left|X^{\left(i\right)}\right|\left|Y_{\left(i\right)}\right|\sum_{l\neq i}\left|X^{\left(l\right)}\right|\left|Y_{\left(l\right)}\right|. \end{array}$$From Equation (Equation 12), Equation (Equation 13), and Equation (Equation 14), we have
(15)$$\begin{array}{c} \left\{f_{1}\left(x,y\right)+f_{2}\left(x,y\right)\right\}-\left\{f_{1}\left(z,y\right)+f_{2}\left(z,y\right)\right\}\geq 0. \end{array}$$
This inequality means that Equation (Equation 8) is minimized at $x_{i}=0$ if all elements of $A^{\prime (i)}$ are zero.    □

If all the elements in $A^{\left(i\right)}$ are identical, then according to the definitions of A and $A^{\prime }$, all elements in $A^{\prime (i)}$ are also identical. Hence, applying Theorem 1, we can deduce that the value of $x_{i}$ yields the minimum value of *E* is 0, which implies $A^{\prime (i)}-x_{i}e_{m}=0$. Combining this equation with Equation (Equation 8), we can obtain the values of $x_{l}$(l≠i), which minimize *E* by extracting the submatrices, respectively, after removing the *i*th column from $A^{\prime }$ and the *i*th row from $B^{\prime }$.

Therefore, to minimize *E*, we can eliminate the *i*th column of $A^{\prime }$ and the *i*th row of $B^{\prime }$ if all of their elements are identical. Subsequently, we can minimize the error *E* using the resulting submatrix, which can be obtained by removing identical elements, as presented in Equation (Equation 8). Hereafter, we denote the standard deviation of the elements in $A^{\prime (i)}$ and $B_{\left(i\right)}^{\prime }$ by $\sigma_{i}$ and $\delta_{i}$, respectively, with the assumption that $\sigma_{i}>0$ and $\delta_{i}>0$ for all *i*.

**Theorem** **2.**
*The following equations hold for any i.*

(16)
$$\begin{array}{c} {\left.\frac{\partial E}{\partial x_{i}}\right|}_{\begin{array}{c} x=0\\ y=0 \end{array}}=0,\;{\left.\frac{\partial E}{\partial y_{i}}\right|}_{\begin{array}{c} x=0\\ y=0 \end{array}}=0. \end{array}$$



**Proof.** Let ${\left[M\right]}_{i,j}$ be the *i*th row and *j*th column element of matrix M. For matrices $A\in R^{m\times k}$ and $B\in R^{k\times n}$, we have the following equalities:
(17)$$\begin{array}{cc} {∥\mathrm{AB}∥}^{2}\; & =\sum_{i=1}^{m}\sum_{j=1}^{n}{\left(\sum_{l=1}^{k}{\left[A\right]}_{i,l}{\left[B\right]}_{l,j}\right)}^{2}\\ \; & =\sum_{i=1}^{m}\sum_{j=1}^{n}\sum_{l_{1},l_{2}}{\left[A\right]}_{i,l_{1}}{\left[A\right]}_{i,l_{2}}{\left[B\right]}_{l_{1},j}{\left[B\right]}_{l_{2},j}\\ \; & =\sum_{l_{1},l_{2}}\sum_{i=1}^{m}\sum_{j=1}^{n}{\left[A\right]}_{i,l_{1}}{\left[A\right]}_{i,l_{2}}{\left[B\right]}_{l_{1},j}{\left[B\right]}_{l_{2},j}\\ \; & =\sum_{l_{1},l_{2}}\left(\sum_{i=1}^{m}{\left[A\right]}_{i,l_{1}}{\left[A\right]}_{i,l_{2}}\right)\left(\sum_{j=1}^{n}{\left[B\right]}_{l_{1},j}{\left[B\right]}_{l_{2},j}\right)=\sum_{l_{1},l_{2}}\left(A^{(l_{1})⊤}A^{\left(l_{2}\right)}\right)\left(B_{\left(l_{1}\right)}B_{\left(l_{2}\right)}^{⊤}\right) \end{array}$$
Using Equation (Equation 17), we can reformulate a term in Equation (Equation 8) as
(18)$$\begin{array}{c} {∥\left(A^{\prime }-e_{m}x^{⊤}\right)\left(B^{\prime }-ye_{n}^{⊤}\right)∥}^{2}=\sum_{l_{1},l_{2}}\left\{{\left(A^{\prime (l_{1})}-x_{l_{1}}e_{m}\right)}^{⊤}\left(A^{\prime (l_{2})}-x_{l_{2}}e_{m}\right)\right\}\left\{\left(B_{\left(l_{1}\right)}^{\prime }-y_{l_{1}}e_{n}^{⊤}\right)\left(B_{\left(l_{2}\right)}^{\prime ⊤}-y_{l_{2}}e_{n}\right)\right\}. \end{array}$$
Since the sum of each column of $A^{\prime }$ and each row of $B^{\prime }$ is zero, $A^{\prime (l)⊤}e_{m}=0$ and $B_{\left(l\right)}^{\prime }e_{n}=0$ are valid for any *l*. Applying this to the equality above, we have
(19)$$\begin{array}{cc} {∥\left(A^{\prime }-e_{m}x^{⊤}\right)\left(B^{\prime }-ye_{n}^{⊤}\right)∥}^{2}\; & =\sum_{l_{1},l_{2}}\left(A^{\prime (l_{1})⊤}A^{\prime (l_{2})}+mx_{l_{1}}x_{l_{2}}\right)\left(B_{\left(l_{1}\right)}^{\prime }B^{\prime ⊤}{}_{\left(l_{2}\right)}+ny_{l_{1}}y_{l_{2}}\right)\\ \; & =∥A^{\prime }B^{\prime }{∥}^{2}+{∥A^{\prime }ye_{n}^{⊤}∥}^{2}+{∥B^{\prime ⊤}xe_{m}^{⊤}∥}^{2}+{∥e_{m}x^{⊤}ye_{n}^{⊤}∥}^{2}\\ \; & =∥A^{\prime }B^{\prime }{∥}^{2}+n{\left|A^{\prime }y\right|}^{2}+m{\left|B^{\prime ⊤}x\right|}^{2}+mn{\left(x^{⊤}y\right)}^{2}. \end{array}$$
Since the average of elements in $A^{\prime (i)}$ is zero, we have $\sigma_{i}^{2}=\left(\sum_{l}{\left[A^{\prime (i)}\right]}_{l}^{2}\right)/m$, where ${\left[v\right]}_{i}$ denotes the *i*th element of vector v. From this equality, we obtain
(20)$$\begin{array}{c} \left|A^{\prime (i)}-x_{i}e_{m}\right|=\sqrt{\sum_{l=1}^{m}{\left({\left[A^{\prime (i)}\right]}_{l}-x_{i}\right)}^{2}}=\sqrt{\sum_{l=1}^{m}\left({\left[A^{\prime (i)}\right]}_{l}^{2}+x_{i}^{2}\right)}=\sqrt{m\left(\sigma_{i}^{2}+x_{i}^{2}\right)}. \end{array}$$
In the same way, we have
(21)$$\begin{array}{c} \;\left|B_{\left(i\right)}^{\prime ⊤}-y_{i}e_{n}\right|=\sqrt{n\left(\delta_{i}^{2}+y_{i}^{2}\right)}. \end{array}$$
Substituting Equation (Equation 19), Equation (Equation 20), and Equation (Equation 21) into Equation (Equation 8), we obtain
(22)$$\begin{array}{c} E\;=\;mn{\left(\sum_{l=1}^{k}\sqrt{\left(\sigma_{l}^{2}+x_{l}^{2}\right)\left(\delta_{l}^{2}+y_{l}^{2}\right)}\right)}^{2}\;-\;{∥A^{\prime }B^{\prime }∥}^{2}\;-\;n{\left|A^{\prime }y\right|}^{2}\;-\;m{\left|B^{\prime ⊤}x\right|}^{2}\;-\;mn{\left(x^{⊤}y\right)}^{2}. \end{array}$$
Because we have ${\left|B^{\prime ⊤}x\right|}^{2}=x^{⊤}B^{\prime }B^{\prime ⊤}x=\sum_{l_{1},l_{2}}{\left[B^{\prime }B^{\prime ⊤}\right]}_{l_{1},l_{2}}x_{l_{1}}x_{l_{2}}$, the following holds.
(23)$$\begin{array}{c} \frac{\partial {\left|B^{\prime ⊤}x\right|}^{2}}{\partial x_{i}}\;=2\sum_{j}{\left[B^{\prime }B^{\prime ⊤}\right]}_{i,j}x_{j}=2{\left[B^{\prime }B^{\prime ⊤}x\right]}_{i} \end{array}$$
Thus,
(24)$$\begin{array}{cc} \frac{\partial E}{\partial x_{i}}\; & =2mnx_{i}\left(\sum_{l=1}^{k}\sqrt{\left(\sigma_{l}^{2}+x_{l}^{2}\right)\left(\delta_{l}^{2}+y_{l}^{2}\right)}\right)\sqrt{\frac{\delta_{i}^{2}+y_{i}^{2}}{\sigma_{i}^{2}+x_{i}^{2}}}-2m{\left[B^{\prime }B^{\prime ⊤}x\right]}_{i}-2mny_{i}x^{⊤}y. \end{array}$$
In the same way, we have
(25)$$\begin{array}{cc} \frac{\partial E}{\partial y_{i}}\; & =2mny_{i}\left(\sum_{l=1}^{k}\sqrt{\left(\sigma_{l}^{2}+x_{l}^{2}\right)\left(\delta_{l}^{2}+y_{l}^{2}\right)}\right)\sqrt{\frac{\sigma_{i}^{2}+x_{i}^{2}}{\delta_{i}^{2}+y_{i}^{2}}}-2n{\left[A^{\prime ⊤}A^{\prime }y\right]}_{i}-2mnx_{i}x^{⊤}y. \end{array}$$
By combining Equation (Equation 24) with Equation (Equation 25), we can derive Equation (Equation 16).    □

**Theorem** **3.**
*The largest eigenvalue of $MM^{⊤}$ is less than or equal to ${∥M∥}^{2}$ for any real nonzero matrix M. The equality holds if the rank of M is one.*


**Proof.** Let $\lambda_{i}$ be the *i*th largest eigenvalue of $MM^{⊤}$. Since $MM^{⊤}$ is positive semidefinite, all eigenvalues of $MM^{⊤}$ are non-negative, and we have ${∥M∥}^{2}=\mathrm{Tr}\left(MM^{⊤}\right)=\sum_{i}\lambda_{i}\geq \lambda_{1}$. Therefore, the largest eigenvalue of $MM^{⊤}$ is less than or equal to ${∥M∥}^{2}$.The equality holds if and only if all eigenvalues other than the largest one are zero, which means that the rank of $MM^{⊤}$ is one. Hereafter, we assume that the rank of $MM^{⊤}$ is one. Let *r* be the rank of $M\in R^{m\times n}$. We then have the singular value decomposition (SVD) of M as $MU\Sigma V^{⊤}$, where $U\in R^{m\times r},\Sigma \in R^{r\times r}$, and $V\in R^{n\times r}$. Because V is an orthogonal matrix and Σ is a diagonal matrix, $MM^{⊤}=U\Sigma^{2}U^{⊤}$ holds. This is nothing but the SVD of $MM^{⊤}$, and it implies that the rank of $MM^{⊤}$ is *r* as well. Since the rank of $MM^{⊤}$ is one, we have r=1; thus, the rank of M is one.    □

**Theorem** **4.**
*The following equation holds for any combination of i and j.*

(26)
$$\begin{array}{c} {\left.\frac{\partial^{2}E}{\partial x_{i}\partial y_{j}}\right|}_{\begin{array}{c} x=0\\ y=0 \end{array}}=0. \end{array}$$



**Proof.** From Equation (Equation 24), we have
(27)$$\begin{array}{cc} \frac{\partial^{2}E}{\partial x_{i}\partial y_{i}}\; & =\frac{2mnx_{i}y_{i}}{\sqrt{\left(\sigma_{i}^{2}+x_{i}^{2}\right)\left(\delta_{i}^{2}+y_{i}^{2}\right)}}\left(\sum_{l=1}^{k}\sqrt{\left(\sigma_{l}^{2}+x_{l}^{2}\right)\left(\delta_{l}^{2}+y_{l}^{2}\right)}\right)-2mnx^{⊤}y. \end{array}$$
Furthermore,
(28)$$\begin{array}{cc} \frac{\partial^{2}E}{\partial x_{i}\partial y_{j}}\; & =2mnx_{i}y_{j}\left(\sum_{l=1}^{k}\sqrt{\left(\sigma_{l}^{2}+x_{l}^{2}\right)\left(\delta_{l}^{2}+y_{l}^{2}\right)}\right)\sqrt{\frac{\delta_{i}^{2}+y_{i}^{2}}{\sigma_{i}^{2}+x_{i}^{2}}\cdot \frac{\sigma_{j}^{2}+x_{j}^{2}}{\delta_{j}^{2}+y_{j}^{2}}}-2mnx_{j}y_{i} \end{array}$$
holds for any i≠j. Therefore, we have Equation (Equation 26).    □

**Theorem** **5.**
*The Hessian matrices of E, regarding x and y, and $\nabla_{xx}^{2}E$ and $\nabla_{yy}^{2}E$, are positive semidefinite at x=y=0.*


**Proof.** We define $f\left(x_{i}\right)\sum_{l=1}^{k}\sqrt{\left(\sigma_{l}^{2}+x_{l}^{2}\right)\left(\delta_{l}^{2}+y_{l}^{2}\right)}$ for simplicity. Then, we have
(29)$$\begin{array}{cc} \frac{\partial }{\partial x_{i}}\left\{x_{i}f\left(x_{i}\right)\sqrt{\frac{\delta_{i}^{2}+y_{i}^{2}}{\sigma_{i}^{2}+x_{i}^{2}}}\right\}\; & =f\left(x_{i}\right)\frac{\partial }{\partial x_{i}}\left(x_{i}\sqrt{\frac{\delta_{i}^{2}+y_{i}^{2}}{\sigma_{i}^{2}+x_{i}^{2}}}\right)+x_{i}\sqrt{\frac{\delta_{i}^{2}+y_{i}^{2}}{\sigma_{i}^{2}+x_{i}^{2}}}\frac{\partial f(x_{i})}{\partial x_{i}}\\ \; & =f\left(x_{i}\right)\frac{\sigma_{i}^{2}\sqrt{\delta_{i}^{2}+y_{i}^{2}}}{{\left(\sigma_{i}^{2}+x_{i}^{2}\right)}^{\frac{3}{2}}}+\frac{x_{i}^{2}\left(\delta_{i}^{2}+y_{i}^{2}\right)}{\sigma_{i}^{2}+x_{i}^{2}}. \end{array}$$
Moreover, the following equalities hold.
(30)$$\begin{array}{c} \left\{\begin{array}{c} \frac{\partial }{\partial x_{i}}{\left[B^{\prime }B^{\prime ⊤}x\right]}_{i}=\frac{\partial }{\partial x_{i}}\sum_{j}{\left[B^{\prime }B^{\prime ⊤}\right]}_{i,j}x_{j}={\left[B^{\prime }B^{\prime ⊤}\right]}_{i,i}\\ \frac{\partial }{\partial x_{i}}\left(y_{i}x^{⊤}y\right)=y_{i}^{2} \end{array}\right. \end{array}$$
Substituting these equalities into Equation (Equation 24), we can simplify $\partial^{2}E/\partial x_{i}^{2}$ as
(31)$$\begin{array}{c} \frac{2mn\sigma_{i}^{2}\sqrt{\delta_{i}^{2}+y_{i}^{2}}}{{\left(\sigma_{i}^{2}+x_{i}^{2}\right)}^{\frac{3}{2}}}\left(\sum_{l=1}^{k}\sqrt{\left(\sigma_{l}^{2}+x_{l}^{2}\right)\left(\delta_{l}^{2}+y_{l}^{2}\right)}\right)+\frac{2mnx_{i}^{2}\left(\delta_{i}^{2}+y_{i}^{2}\right)}{\sigma_{i}^{2}+x_{i}^{2}}-2m{\left[B^{\prime }B^{\prime ⊤}\right]}_{i,i}-2mny_{i}^{2}. \end{array}$$
Since ${\left[B^{\prime }B^{\prime ⊤}\right]}_{i,i}$ is equal to the square of the Euclidean length of $B_{\left(i\right)}^{\prime }$, we have ${\left[B^{\prime }B^{\prime ⊤}\right]}_{i,i}=n\delta_{i}^{2}$. Substituting this into Equation (Equation 31), we obtain
(32)$$\begin{array}{c} \frac{\partial^{2}E}{\partial x_{i}^{2}}=\frac{2mn\sigma_{i}^{2}}{\sigma_{i}^{2}+x_{i}^{2}}\sqrt{\frac{\delta_{i}^{2}+y_{i}^{2}}{\sigma_{i}^{2}+x_{i}^{2}}}\sum_{l\neq i}\sqrt{\left(\sigma_{l}^{2}+x_{l}^{2}\right)\left(\delta_{l}^{2}+y_{l}^{2}\right)}. \end{array}$$
This equation implies that
(33)$$\begin{array}{c} {\left.\frac{\partial^{2}E}{\partial x_{i}^{2}}\right|}_{\begin{array}{c} x=0\\ y=0 \end{array}}=\frac{2mn\delta_{i}}{\sigma_{i}}\sum_{l\neq i}\sigma_{l}\delta_{l}. \end{array}$$
For i≠j, we can simplify $\partial^{2}E/\partial x_{i}\partial x_{j}$ as
(34)$$\begin{array}{c} 2mn\sqrt{\frac{\delta_{i}^{2}+y_{i}^{2}}{\sigma_{i}^{2}+x_{i}^{2}}\cdot \frac{\delta_{j}^{2}+y_{j}^{2}}{\sigma_{j}^{2}+x_{j}^{2}}}x_{i}x_{j}-2m{\left[B^{\prime }B^{\prime ⊤}\right]}_{i,j}-2mny_{i}y_{j}. \end{array}$$
This equation implies that
(35)$$\begin{array}{c} {\left.\frac{\partial^{2}E}{\partial x_{i}\partial x_{j}}\right|}_{\begin{array}{c} x=0\\ y=0 \end{array}}=-2m{\left[B^{\prime }B^{\prime ⊤}\right]}_{i,j}. \end{array}$$Combining Equations (Equation 33) and (Equation 35), we can express the Hessian matrix of *E* regarding x at x=y=0 as follows:
(36)$$\begin{array}{cc} {\left.\nabla_{xx}^{2}E\right|}_{\begin{array}{c} x=0\\ y=0 \end{array}}:=\;\;\left[\begin{array}{cccc} \frac{\partial^{2}E}{\partial x_{1}^{2}} & \frac{\partial^{2}E}{\partial x_{1}\partial x_{2}} & \ldots & \frac{\partial^{2}E}{\partial x_{1}\partial x_{k}}\\ \frac{\partial^{2}E}{\partial x_{1}\partial x_{2}} & \frac{\partial^{2}E}{\partial x_{2}^{2}} & \ldots & \frac{\partial^{2}E}{\partial x_{2}\partial x_{k}}\\ \vdots & \vdots & \ddots & \vdots \\ \frac{\partial^{2}E}{\partial x_{1}\partial x_{k}} & \frac{\partial^{2}E}{\partial x_{2}\partial x_{k}} & \ldots & \frac{\partial^{2}E}{\partial x_{k}^{2}} \end{array}\right]\;\; & =\;2m\;\left\{n\;\left(\sum_{l=1}^{k}\sigma_{l}\delta_{l}\right)\;\left[\begin{array}{ccc} \frac{\delta_{1}}{\sigma_{1}} & \; & O\\ \; & \ddots & \;\\ O & \; & \frac{\delta_{k}}{\sigma_{k}} \end{array}\right]\;-\;B^{\prime }B^{\prime ⊤}\right\} \end{array}$$Let Z be a diagonal and regular matrix $Z\mathrm{diag}(\sqrt{\delta_{1}/\sigma_{1}},\ldots ,\sqrt{\delta_{k}/\sigma_{k}})$. From Theorem 3, the largest eigenvalue of $Z^{-1}B^{\prime }B^{\prime ⊤}{\left(Z^{-1}\right)}^{⊤}$ is less than or equal to ${∥Z^{-1}B^{\prime }∥}^{2}=\sum_{l=1}^{k}{∥\sqrt{\sigma_{l}/\delta_{l}}B_{\left(l\right)}^{\prime }∥}^{2}=n\sum_{l=1}^{k}\sigma_{l}\delta_{l}.$ Thus,
(37)$$\begin{array}{c} n\left(\sum_{l=1}^{k}\sigma_{l}\delta_{l}\right)I-Z^{-1}B^{\prime }B^{\prime ⊤}{\left(Z^{-1}\right)}^{⊤}⪰O, \end{array}$$
where I is an identity matrix. Therefore, $nZ\left(\sum_{l=1}^{k}\sigma_{l}\delta_{l}\right)Z-B^{\prime }B^{⊤}⪰O$ holds, which shows that $\nabla_{\mathrm{xx}}^{2}E$ is positive semidefinite at x=y=0. In the same way, we can prove that $\nabla_{\mathrm{yy}}^{2}E$ is positive semidefinite at x=y=0.    □

**Theorem** **6.**
*The function E is locally optimal at x=y=0.*


**Proof.** The Hessian matrix of *E* can be expressed as
(38)$$\begin{array}{c} \nabla^{2}E:=\left[\begin{array}{cc} \nabla_{\mathrm{xx}}^{2}E & \nabla_{\mathrm{xy}}^{2}E\\ \nabla_{\mathrm{yx}}^{2}E & \nabla_{\mathrm{yy}}^{2}E \end{array}\right]. \end{array}$$
Theorem 4 provides $\nabla_{\mathrm{xy}}^{2}E=\nabla_{\mathrm{yx}}^{2}E=O$ at x=y=0. Furthermore, Theorem 5 claims that $\nabla_{\mathrm{xx}}^{2}E$ and $\nabla_{\mathrm{yy}}^{2}E$ are positive semidefinite at x=y=0, which means $\nabla^{2}E$ is also positive semidefinite at x=y=0. Based on the positive semidefinite property and Theorem 2, it is apparent that the point x=y=0 serves as a locally optimal solution for the minimization of *E*.    □

The preceding discussion demonstrates that the point x=y=0 constitutes a local optimum of *E*. Hereafter, we establish that, in almost all cases, this point is a global optimum because *E* has only one local optimum, and, in other cases, the point is globally optimal as well because all local optima are connected.

**Theorem** **7.**
*Let M be a rank-1 matrix and v be a vector satisfying $MM^{⊤}=vv^{⊤}$. The vector v is the only eigenvector of $MM^{⊤}$. The matrix ${∥M∥}^{2}I-MM^{⊤}$ has an eigenvalue of 0, and only a scaled version of v is its eigenvector.*


**Proof.** As ${∥M∥}^{2}=\mathrm{Tr}\left(MM^{⊤}\right)=\mathrm{Tr}\left(vv^{⊤}\right)={\left|v\right|}^{2}$ holds, we have $\left({∥M∥}^{2}I-MM^{⊤}\right)v={\left|v\right|}^{2}v-vv^{⊤}v=0$. Since M is a rank-1 matrix, $MM^{⊤}$ is a rank-1 matrix as well and has only one eigenvalue. Combining this and ${∥M∥}^{2}v=MM^{⊤}v$, we can see that v is the unique eigenvector of $MM^{⊤}$. Moreover, based on the equality above, the matrix ${∥M∥}^{2}I-MM^{⊤}$ has an eigenvalue of 0, and only a scaled version of v becomes its eigenvector.    □

**Theorem** **8.**
*Let M be a rank-1 matrix, and v be a vector satisfying $MM^{⊤}=vv^{⊤}$. Let Z be a regular and diagonal matrix. The vectors x that satisfy $x^{⊤}\left({∥Z^{-1}M∥}^{2}Z^{2}-MM^{⊤}\right)x=0$ are scaled versions of $Z^{-2}v$.*


**Proof.** Since M is a rank-1 matrix and Z is regular, $Z^{-1}M$ is a rank-1 matrix as well. Let Q be ${∥Z^{-1}M∥}^{2}I-\left(Z^{-1}M\right){\left(Z^{-1}M\right)}^{⊤}$, and $\tilde{v}$ be a vector satisfying $\tilde{v}{\tilde{v}}^{⊤}=\left(Z^{-1}M\right){\left(Z^{-1}M\right)}^{⊤}$. Theorem 7 implies that Q has an eigenvalue of 0 and only a scaled version of $\tilde{v}$ becomes its eigenvector. Because $\left(Z\tilde{v}\right){\left(Z\tilde{v}\right)}^{⊤}=MM^{⊤}$ holds, we have $v=Z\tilde{v}$. Thus, a scaled version of $Z^{-1}v$ is an eigenvector of Q corresponding to an eigenvalue of 0. Therefore, the following equality holds for any real value of *c*.
(39)$$\begin{array}{c} Q\left(cZ^{-1}v\right)=0 \end{array}$$
We can easily see that $\mathrm{ZQZ}\left(cZ^{-2}v\right)=0$ holds, and it implies that the matrix ZQZ has the eigenvalue of 0 and only a scaled version of $Z^{-2}v$ is one of its eigenvectors. Moreover, we have the following equalities:
(40)$$\begin{array}{c} \mathrm{ZQZ}=Z\left({∥Z^{-1}M∥}^{2}I-Z^{-1}MM^{⊤}{\left(Z^{-1}\right)}^{⊤}\right)Z={∥Z^{-1}M∥}^{2}Z^{2}-MM^{⊤} \end{array}$$
Therefore, the matrix ${∥Z^{-1}M∥}^{2}Z^{2}-MM^{⊤}$ has an eigenvalue of 0, and a scaled version of $Z^{-2}v$ is one of its eigenvectors.Theorem 3 claims that Q is positive semidefinite, and it implies that ZQZ is positive semidefinite as well. Thus, the vector x satisfying $x^{⊤}\left({∥Z^{-1}M∥}^{2}Z^{2}-MM^{⊤}\right)x=0$ is zero vector or a scaled version of the eigenvector corresponding to the eigenvalue of 0. Therefore, x are scaled versions of $Z^{-2}v$.    □

**Theorem** **9.***The function E is globally optimal at*  *x* = *y* = 0.

**Proof.** From Equation (Equation 24), we obtain ∇_*x*_*E* as

(41)$$\begin{array}{c} 2mn\left(\sum_{l=1}^{k}\sqrt{\left(\sigma_{l}^{2}+x_{l}^{2}\right)\left(\delta_{l}^{2}+y_{l}^{2}\right)}\right)\left[\begin{array}{ccc} \sqrt{\frac{\delta_{1}^{2}+y_{1}^{2}}{\sigma_{1}^{2}+x_{1}^{2}}} & \; & O\\ \; & \ddots & \\ O & \; & \sqrt{\frac{\delta_{k}^{2}+y_{k}^{2}}{\sigma_{k}^{2}+x_{k}^{2}}} \end{array}\right]x-2mB^{\prime }B^{\prime ⊤}x-2mn\left(x^{⊤}y\right)y. \end{array}$$
This implies that
(42)$$\begin{array}{cc} \frac{x^{⊤}\nabla_{x}E}{2m}\; & \;=\;x^{⊤}\;\left\{n\left(\;\sum_{l=1}^{k}\sqrt{\left(\sigma_{l}^{2}+x_{l}^{2}\right)\left(\delta_{l}^{2}+y_{l}^{2}\right)}\;\right)\;\;\left[\begin{array}{ccc} \sqrt{\frac{\delta_{1}^{2}+y_{1}^{2}}{\sigma_{1}^{2}+x_{1}^{2}}} & \; & O\\ \; & \ddots & \\ O & \; & \sqrt{\frac{\delta_{k}^{2}+y_{k}^{2}}{\sigma_{k}^{2}+x_{k}^{2}}} \end{array}\right]\;\;-\;\left(\;B^{\prime }B^{\prime ⊤}\;+\;nyy^{⊤}\;\right)\;\;\right\}x. \end{array}$$
We study the conditions that x must satisfy in the local solution of *E*. In that case, $x^{⊤}\nabla_{x}E$ must be zero due to $\nabla_{x}E=0$.

Let $B^{\prime \prime }$ be $B^{\prime \prime }B^{\prime },\sqrt{n}y\in R^{k\times (m+1)}$ and $Z^{\prime }$ be a diagonal and regular matrix
(43)$$\begin{array}{c} Z^{\prime }:=\mathrm{diag}\left({\left(\frac{\delta_{1}^{2}+y_{1}^{2}}{\sigma_{1}^{2}+x_{1}^{2}}\right)}^{\frac{1}{4}},\ldots ,{\left(\frac{\delta_{k}^{2}+y_{k}^{2}}{\sigma_{k}^{2}+x_{k}^{2}}\right)}^{\frac{1}{4}}\right). \end{array}$$
From Theorem 3, the largest eigenvalue of $Z^{\prime -1}B^{\prime \prime }B^{\prime \prime ⊤}{\left(Z^{\prime -1}\right)}^{⊤}$ is less than or equal to
(44)$$\begin{array}{cc} {∥Z^{\prime -1}B^{\prime \prime }∥}^{2}\; & =\sum_{l=1}^{k}{\left|{\left(\frac{\sigma_{l}^{2}+x_{l}^{2}}{\delta_{l}^{2}+y_{l}^{2}}\right)}^{\frac{1}{4}}B_{\left(l\right)}^{\prime \prime }\right|}^{2}\\ \; & =\sum_{l=1}^{k}\sqrt{\frac{\sigma_{l}^{2}+x_{l}^{2}}{\delta_{l}^{2}+y_{l}^{2}}}{\left|\left[\begin{array}{c} B_{\left(l\right)}^{\prime ⊤}\\ \sqrt{n}y_{l} \end{array}\right]\right|}^{2}\\ \; & =\sum_{l=1}^{k}\sqrt{\frac{\sigma_{l}^{2}+x_{l}^{2}}{\delta_{l}^{2}+y_{l}^{2}}}n\left(\delta_{l}^{2}+y_{l}^{2}\right)=n\sum_{l=1}^{k}\sqrt{\left(\sigma_{l}^{2}+x_{l}^{2}\right)\left(\delta_{l}^{2}+y_{l}^{2}\right)}. \end{array}$$
Thus, we have
(45)$$\begin{array}{c} n\sum_{l=1}^{k}\sqrt{\left(\sigma_{l}^{2}+x_{l}^{2}\right)\left(\delta_{l}^{2}+y_{l}^{2}\right)}I_{k}-Z^{\prime -1}B^{\prime \prime }B^{\prime \prime ⊤}{\left(Z^{\prime -1}\right)}^{⊤}⪰O. \end{array}$$
In particular, "≻" holds if $\mathrm{rank}(B^{\prime \prime })>1$.

(i)$\mathrm{rank}(B^{\prime })>1$, or $\mathrm{rank}(B^{\prime })=1$ and y is not a scaled version of a column vector of $B^{\prime }$

Because the rank of $B^{\prime \prime }$ is greater than 1, we have $n\sum_{l=1}^{k}\sqrt{\left(\sigma_{l}^{2}+x_{l}^{2}\right)\left(\delta_{l}^{2}+y_{l}^{2}\right)}Z^{\prime }Z^{\prime ⊤}-B^{\prime \prime }B^{\prime \prime ⊤}≻O$. Combining this inequality and Equation (Equation 42), we see that $x^{⊤}\nabla_{x}E=0$ holds at x=0 only. Therefore, if x is a local minimum of *E*, then $\nabla_{x}E=0$ holds and we have x=0. Since a local minimum is limited to x=0, it also becomes the global minimum.

(ii)$\mathrm{rank}(B^{\prime })=1$ and y is a scaled version of a column vector of $B^{\prime }$

All rows of $B^{\prime }$ are dependent due to $\mathrm{rank}(B^{\prime })=1$. As y is a scaled version of a column vector of $B^{\prime }$, we can express $B^{\prime }$ as $B^{\prime }={y_{1}u,\ldots ,y_{k}u}^{⊤}$ by introducing a vector u. By definition, $y_{i}$ is not always zero.

Substituting Equation (Equation 44) into Equation (Equation 42), we have
(46)$$\begin{array}{c} \frac{x^{⊤}\nabla_{x}E}{2m}=x^{⊤}\left(∥Z^{-1}B^{\prime \prime }∥Z^{\prime 2}-B^{\prime \prime }B^{\prime \prime ⊤}\right)x. \end{array}$$
The right-hand side of Equation (Equation 46) is zero due to $x^{⊤}\nabla_{x}E=0$. Theorem 8 claims that the vectors x satisfying $x^{⊤}\left(∥Z^{-1}B^{\prime \prime }∥Z^{\prime 2}-B^{\prime \prime }B^{\prime \prime ⊤}\right)x=0$ are scaled versions of $Z^{\prime -2}v$, where $vv^{⊤}=B^{\prime \prime }B^{\prime \prime ⊤}$. The equality $B^{\prime \prime }=yu^{⊤},\sqrt{n}$ gives us $B^{\prime \prime }B^{\prime \prime ⊤}=\left({\left|u\right|}^{2}+n\right)yy^{⊤}$, and it means $v=\sqrt{{\left|u\right|}^{2}+n}y$. Thus,
(47)$$\begin{array}{c} x_{i}=c\frac{\sqrt{\sigma_{i}^{2}+x_{i}^{2}}}{\sqrt{\delta_{i}^{2}+y_{i}^{2}}}\sqrt{{\left|u\right|}^{2}+n}y_{i}=c\frac{\sqrt{\sigma_{i}^{2}+x_{i}^{2}}}{\sqrt{y_{i}^{2}{\left|u\right|}^{2}+y_{i}^{2}}}\sqrt{{\left|u\right|}^{2}+n}y_{i}=c^{\prime }\sqrt{\sigma_{i}^{2}+x_{i}^{2}} \end{array}$$
must be satisfied with a certain real value of *c*, where $c^{\prime }:=c\sqrt{\left({\left|u\right|}^{2}+n\right)/{\left(|u\right|}^{2}+1)}$. Solving this equation, we have $x_{i}=\pm c^{\prime }\sigma_{i}/\sqrt{1-c^{\prime 2}}$. This means that if x is a scaled version of vector ${\left[\sigma_{1},\ldots ,\sigma_{k}\right]}^{⊤}$ then we have $\nabla_{x}E=0$. Because x can be 0, points satisfying $\nabla_{x}E=0$ have the same value of *E* at x=0.

We show that $\nabla_{\mathrm{xx}}^{2}E⪰O$ if x is a scaled version of vector ${\left[\sigma_{1},\ldots ,\sigma_{k}\right]}^{⊤}$. Substituting $\sigma_{i}^{2}/\left(\sigma_{i}^{2}+x_{i}^{2}\right)=1-c^{\prime 2}$ into Equation (Equation 32), we have
(48)$$\begin{array}{cc} \frac{1}{2mn}\frac{\partial^{2}E}{\partial x_{i}^{2}}\; & =\left(1-c^{\prime 2}\right)\left(\sqrt{\frac{\delta_{i}^{2}+y_{i}^{2}}{\sigma_{i}^{2}+x_{i}^{2}}}\sum_{l}\sqrt{\left(\sigma_{l}^{2}+x_{l}^{2}\right)\left(\delta_{l}^{2}+y_{l}^{2}\right)}-\left(\delta_{i}^{2}+y_{i}^{2}\right)\right). \end{array}$$
We also obtain the following equality for i≠j by applying $x_{i}^{2}/\left(\sigma_{i}^{2}+x_{i}^{2}\right)=c^{\prime 2}$ into Equation (Equation 34).
(49)$$\begin{array}{cc} \frac{1}{2mn}\frac{\partial^{2}E}{\partial x_{i}\partial x_{j}}\; & =c^{\prime 2}\sqrt{\left(\delta_{i}^{2}+y_{i}^{2}\right)\left(\delta_{j}^{2}+y_{j}^{2}\right)}-\frac{{\left[B^{\prime }B^{\prime ⊤}\right]}_{i,j}+ny_{i}y_{j}}{n}. \end{array}$$
Summarizing Equations (Equation 48), (Equation 49), and the equality ${\left[B^{\prime }B^{\prime ⊤}\right]}_{i,i}+ny_{i}^{2}=n\left(\delta_{i}^{2}+y_{i}^{2}\right)$, we can simplify $\nabla_{\mathrm{xx}}^{2}E$ as follows:
(50)$$\begin{array}{cc} \frac{\nabla_{\mathrm{xx}}^{2}E}{2mn}= & \left(1-c^{\prime 2}\right)\left\{\left(\sum_{l}\sqrt{\left(\sigma_{l}^{2}+x_{l}^{2}\right)\left(\delta_{l}^{2}+y_{l}^{2}\right)}\right)\left[\begin{array}{ccc} \sqrt{\frac{\delta_{1}^{2}+y_{1}^{2}}{\sigma_{1}^{2}+x_{1}^{2}}} & \; & O\\ \; & \ddots & \\ O & \; & \sqrt{\frac{\delta_{k}^{2}+y_{k}^{2}}{\sigma_{k}^{2}+x_{k}^{2}}} \end{array}\right]-\frac{B^{\prime \prime }B^{\prime \prime ⊤}}{n}\right\}\\ \; & +c^{\prime 2}\left\{\left[\begin{array}{c} \sqrt{\delta_{1}^{2}+y_{1}^{2}}\\ \vdots \\ \sqrt{\delta_{k}^{2}+y_{k}^{2}} \end{array}\right]\left[\begin{array}{ccc} \sqrt{\delta_{1}^{2}+y_{1}^{2}} & \cdots & \sqrt{\delta_{k}^{2}+y_{k}^{2}} \end{array}\right]-\frac{B^{\prime \prime }B^{\prime \prime ⊤}}{n}\right\} \end{array}$$
The assumption on $B^{\prime }$ and definition of $\delta_{i}$ attain $\sqrt{\delta_{i}^{2}+y_{i}^{2}}=y_{i}\sqrt{{\left|u\right|}^{2}+1}$ for all *i*. Thus, we have
(51)$$\begin{array}{cc} \left[\begin{array}{c} \sqrt{\delta_{1}^{2}+y_{1}^{2}}\\ \vdots \\ \sqrt{\delta_{k}^{2}+y_{k}^{2}} \end{array}\right]\left[\begin{array}{ccc} \sqrt{\delta_{1}^{2}+y_{1}^{2}} & \cdots & \sqrt{\delta_{k}^{2}+y_{k}^{2}} \end{array}\right]-\frac{B^{\prime \prime }B^{\prime \prime ⊤}}{n}\; & =\left(1-\frac{1}{n}\right){\left|u\right|}^{2}yy^{⊤}⪰O. \end{array}$$
The discussion regarding the positive semidefiniteness of Equation (Equation 36) also demonstrates that the first term on the right-hand side of Equation (Equation 50) is positive semidefinite. Therefore, $\nabla_{\mathrm{xx}}^{2}E⪰O$ holds if x is a scaled version of vector ${\left[\sigma_{1},\cdots ,\sigma_{k}\right]}^{⊤}$.

Through the discussion on the conditions under which x must satisfy in the local solution of *E* when y is a constant vector, the following revelations emerge:

If x is a local solution, it must be a scaled version of vector ${\left[\sigma_{1},\cdots ,\sigma_{k}\right]}^{⊤}$.All local solutions are continuous and have the same value of *E*.$\nabla_{\mathrm{xx}}^{2}E⪰O$ holds if x is a scaled version of vector ${\left[\sigma_{1},\cdots ,\sigma_{k}\right]}^{⊤}$.

In other words, the set of local solutions is globally minimal, and x=0 also becomes a global minimum.

Both discussions (i) and (ii) yield the conclusion that x=0 is valid when *E* is minimized under the condition of a constant y. Similar observations can be made regarding investigating the conditions for y in the local solution of *E*. Hence, the function *E* attains global optimality at x=y=0.    □

## 3. Results

### 3.1. Assessing the Accuracy of Approximate Matrix Multiplications

We evaluate the approximation performance of the proposed method for matrix multiplication. First, we set the size of the matrices to $m=n=k=2^{15}$. Then, we randomly determine each element of the matrices independently of a uniform distribution between 0 and 1.

Figure 1A evaluates the speedup ratio of the computation time and the approximation performance obtained by applying the conventional and proposed methods for AMM. The horizontal axis compares the computation time of the approximate calculation with that of the exact matrix product calculation. In the experiments of this chapter, we implemented the CUDA program for AMM using CUDA 12.0, which was executed on an NVIDIA A100 GPU. The exact matrix product was computed using an API incorporated into cuBLAS (NVIDIA’s official matrix library). The numerical precision was set to a single-precision floating point. The vertical axis in this figure represents the median relative accuracy between an approximate matrix Z and the exact solution AB, calculated by executing each method 100 times. The relative accuracy is defined as
(52)$$\begin{array}{c} 1-\frac{∥\mathrm{AB}-Z∥}{∥\mathrm{AB}∥}. \end{array}$$

Figure 1A demonstrates that the proposed method achieves higher acceleration while maintaining relative accuracy. For instance, at a relative accuracy of 0.95, the acceleration rate is 87.2 times faster for the conventional method and 443.2 times faster for the proposed method. Although the relative accuracy required depends on the application, the proposed method outperforms the traditional method at all levels of relative accuracy.

Figure 1B displays a graph showing the speedup ratio trend achieving a relative accuracy of 0.95 when the matrix size is changed from $2^{10}$ to $2^{15}$. This demonstrates that the proposed method can achieve higher speedup ratios than the conventional method regardless of the matrix size. Furthermore, the difference in the speedup ratio expands as the matrix size increases. If the matrix size is small, the exact product does not generally become the bottleneck of the entire calculation. The approximate product is required mainly for large-scale computations. The property of the proposed method (that the advantage becomes more pronounced as the matrix size increases) is desirable.

### 3.2. Application to a Practical Problem: Computing Eigenvalues and Eigenvectors

Next, we evaluate the performance of the proposed methodology in practical applications. We focus on the computation of eigenvalues, particularly the largest eigenvalue. Several algorithms have been proposed for computing the largest eigenvalue, including power iteration, inverse power iteration [33], accelerated stochastic power iteration [34], and the LOBPCG method [35]. This paper applies the proposed AMM to the power iteration, which forms the backbone of many existing techniques.

We assume for simplicity that the matrix $A\in R^{m\times m}$ is symmetric. Then, all eigenvalues of A are real. See Algorithm 3. For any square matrix A, let λ be an eigenvalue of A. Then, $A^{2}$ has an eigenvalue of $\lambda^{2}$. Therefore, we substitute A with $A^{2}$ in Algorithm 3.
**Algorithm 3:**  Power method**Require:**  $A\in R^{m\times m}$, x≠0 : *m*-dimensional randomized unit vector.**Ensure:**  λ: the eigenvalue of A with the largest absolute value  1:  **for** while not converged **do**  2:    y←Ax  3:    $\lambda \leftarrow x^{⊤}y$  4:    x←y/∥y∥  5:  **end for**  6:  Return λ

In this case, the second line is expressed as $y\leftarrow A^{2}x$, and we apply the proposed AMM to this step. From Equation (Equation 7), we have
(53)$$\begin{array}{cc} A^{2}\; & =AA^{⊤}=A^{\prime }A^{\prime ⊤}+\left(A\alpha \right)e_{m}^{⊤}+e_{m}{\left(A\alpha \right)}^{⊤}-e_{m}\left(\alpha^{⊤}\alpha \right)e_{m}^{⊤}. \end{array}$$
We execute the calculation of $A^{\prime }A^{\prime ⊤}$ in Equation (Equation 53) using Algorithm 2. By incorporating this AMM into Algorithm 3, we obtain Algorithm 4, a power iteration algorithm that utilizes the approximation. The fourth to eighth lines of Algorithm 4 correspond to the Monte Carlo AMM with the proposed method.
**Algorithm 4:** Power method with the proposed Monte Carlo AMM**Require: **A:  m×m symmetric positive-semidefinite matrix, x≠0: *m*-dimensional randomized unit vector, c∈N**Ensure:**  λ: the largest absolute eigenvalue of A  1:  z←Aα, $v\leftarrow z-{\left|\alpha \right|}^{2}e_{m}$  2:  Set $p_{1},\ldots ,p_{m}$, such that $\sum_{l=1}^{m}p_{l}=1$  3:  **for** while not converged **do**  4:    $y\leftarrow \left(e_{m}^{⊤}x\right)v+\left(z^{⊤}x\right)e_{m}$  5:    **for** t=1,…,c **do**  6:        Pick $i_{t}$ with $P\left[i_{t}=l\right]=p_{l}$  7:        $y\leftarrow y+\frac{A^{\prime (i_{t})}A_{\left(i_{t}\right)}^{\prime ⊤}x}{cp_{i_{t}}}$  8:    **end for**  9:    $\lambda \leftarrow x^{⊤}y$  10:    x←y/∥y∥  11:  **end for**  12:  Return $\sqrt{\lambda }$

In this section, we assess the performance of the proposed methodology by employing the principal component analysis (PCA) [36] as a representative example, which serves as one of the applications for computing eigenvalues. The PCA is a method employed to examine and reduce the dimensionality of the dataset. It involves identifying the principal components and linear combinations of the original features that effectively capture the most substantial variations within the data. The eigenvalues and eigenvectors of a covariance matrix hold a pivotal role in PCA. When presented with a dataset comprising *p* features, the initial step in PCA involves the computation of the covariance matrix. This matrix assumes a symmetric structure with dimensions of *p*-by-*p*, where each element denotes the covariance between two specific features. The quest for the most dominant principal components is tantamount to identifying the largest eigenvector within the covariance matrix. Hence, within this section, we focus on image analysis, an area in which PCA finds frequent application. Specifically, we have obtained the largest eigenvalues and corresponding eigenvectors from a matrix with a size of 11750×11750, which is acquired via the eigenface algorithm [37] on the LFW dataset [38].

Figure 2A illustrates the variation in convergence over time when CUDA programs of the power method (PM) are run on an NVIDIA A100 GPU. Both figures share a common horizontal axis representing the computation time.

The vertical axis in Figure 2A represents the relative error between the largest and estimated eigenvalue during iterations. In contrast, the vertical axis in Figure 2B represents the deviation between the eigenvector corresponding to the largest eigenvalue and the estimated eigenvector during iterations. Incorporating the proposed AMM into the power method requires the execution of the first line of Algorithm 4, resulting in a longer duration until the completion of the initial iteration compared to the conventional method incorporating AMM in the power method. However, since this computation only needs to be performed once, its impact on the overall computation time is limited. Compared to the power method with the conventional AMM, the method with the proposed AMM demonstrates favorable convergence in both indicators. The power method with the proposed AMM cannot achieve the same level of convergence as the one employing exact matrix multiplication. This is because the expected variance from the AMM breaks the condition revealed in previous studies on the noisy power method [39].

To solve this problem, we propose a variance reduction method that performs exact matrix multiplication under some criteria. When obtaining the approximate matrix product of the matrices A and $B+\tilde{B}$, if C=AB is known, we can compute the approximate solution $\tilde{C}\approx A\tilde{B}$ and provide $C+\tilde{C}$ as an approximate solution. As mentioned in Section 2, the expected error will decrease if the Frobenius norm of $\tilde{B}$ is small. The estimated eigenvector converges gradually through the power method. Therefore, computing the exact matrix product only sometimes and minimizing the expected error of the AMM are expected to be effective. Based on this motivation, we present Algorithm 5. It uses the proposed AMM while performing exact matrix multiplication only for the steps that satisfy the given conditions.
**Algorithm 5:** Variance reduction power method with the proposed Monte Carlo AMM**Require:** A:  m×m symmetric positive-semidefinite matrix, x≠0: *m*-dimensional randomized unit vector, c∈N**Ensure:**  λ: the largest absolute eigenvalue of A  1:  z←Aα, $v\leftarrow z-{\left|\alpha \right|}^{2}e_{m}$, $\bar{x}\leftarrow 0$, $\bar{y}\leftarrow 0$  2:  Set $p_{1},\ldots ,p_{m}$, such that $\sum_{l=1}^{m}p_{l}=1$  3:  **for** while not converged **do**  4:    **if** criteria are met **then**  5:        $y\leftarrow \bar{y}+\left(e_{m}^{⊤}\left(x-\bar{x}\right)\right)v+\left(z^{⊤}\left(x-\bar{x}\right)\right)e_{m}$  6:        **for** t=1,…,c **do**  7:              Pick $i_{t}$ with $P\left[i_{t}=l\right]=p_{l}$  8:           $y\leftarrow y+\frac{A^{\prime (i_{t})}A_{\left(i_{t}\right)}^{\prime ⊤}\left(x-\bar{x}\right)}{cp_{i_{t}}}$  9:           **end for**  10:    **else**  11:        $y\leftarrow A^{2}x$, $\bar{x}\leftarrow x$, $\bar{y}\leftarrow y$  12:    **end if**  13:    $\lambda \leftarrow x^{⊤}y$  14:    x←y/∥y∥  15:  **end for**  16:  Return $\sqrt{\lambda }$

Various conditions can be considered for performing exact matrix multiplication. For instance, we can employ a static strategy where an exact computation is performed every few steps. Alternatively, a dynamic strategy is also promising, where convergence stagnation due to approximation errors is detected, and an exact computation is executed. In this section, we adopt a hybrid approach that combines an AMM with the exact computation based on this dynamic strategy. To assess the convergence of eigenvector calculations, we adopt the following rules:1.If the dot product between the current and previous vectors of x exceeds a threshold value θ, then we execute the exact matrix product at the next step.2.Even if the initial condition is met, the subsequent calculation employs an AMM if the exact matrix multiplication was previously executed.

The experimental results are presented in Figure 3. We set the number of samplings *c* to 512. We executed each method 1000 times while changing random seeds. In this figure, the markers represent the median, while the shaded region represents the interquartile range from the first to the third quartiles. The top row corresponds to $\theta =1-10^{-4}$, while the bottom row corresponds to $\theta =1-10^{-3}$. The left panel depicts the relative error between the true largest eigenvalue $\lambda_{\max}$ and the estimated largest eigenvalue ${\tilde{\lambda }}_{\max}$. The right panel illustrates the discrepancy between the eigenvectors corresponding to the largest eigenvalue, denoted as $v_{\max}$, and the estimated eigenvectors denoted as ${\tilde{v}}_{\max}$. The threshold is a hyperparameter, and it is not clear how to determine an appropriate value for it. Therefore, the method should exhibit stable convergence for various threshold values. For $\theta =1-10^{-4}$, the power method with the conventional AMM fails to converge. In contrast, the power method with the proposed AMM demonstrates convergence performance comparable to the power method employing exact matrix multiplication while achieving a reduced computation time. When the threshold is changed to $\theta =1-10^{-3}$, the steps involving exact matrix multiplication increase. Consequently, even the power method with the conventional AMM exhibits good convergence. However, the increased frequency of exact matrix multiplications requires more computation time than the power method with the proposed AMM. Thus, the power method with the proposed AMM demonstrates more robust convergence than the one utilizing only exact matrix multiplication. Moreover, it is faster than the power method using only exact matrix multiplication. For instance, when $\theta =1-10^{-3}$, the computation time until reaching $1-{\left(v_{\max}^{⊤}{\tilde{v}}_{\max}\right)}^{2}=10^{-11}$ is reduced by about 40%.

## 4. Discussion

One of the challenges of the proposed method is that its effectiveness diminishes when the elements of matrix A’s columns, representing α, and the elements of matrix B’s rows, representing β, are close to zero. In practical applications and datasets, matrices often exhibit element distributions with specific tendencies or possess the property of being positive semidefinite, thereby deviating from the scenarios where the proposed method is less effective. However, there are cases in specific applications where these conditions apply, requiring alternative methods to improve the approximation accuracy of the matrices.

In this paper, we applied AMM to the simple power method and computed the largest eigenvalue. By incorporating QR decomposition into the iterations of the power method, we can obtain the first several singular eigenvectors [39]. Furthermore, by extending the application of AMM to other variants of the power method discussed in this paper, we anticipate accelerating the computation of various eigenvalue calculations. As mentioned in the previous section, some prior studies on the noisy power method analyze the convergence behavior when noise is introduced in each iteration of the power method. Although we determined the threshold θ in this paper, it is possible to analytically or dynamically adjust θ based on such prior research. This remains a challenge for future investigation.

Besides the power method, least square minimization can also be considered among the applications of AMM. Previous research has explored accelerating the analysis of large-scale single nucleotide polymorphisms by utilizing approximate calculations to reduce computational complexity, as the use of pseudo-inverse matrices for obtaining approximate solutions incur high computational costs [19]. By applying our proposed concept to randomized numerical linear algebra (RandNLA), including these applications, there is potential to achieve faster computations and obtain higher-quality approximate solutions. Therefore, applying our method to various scientific and technological computations will be important.

## Figures and Tables

**Figure 1 entropy-25-01130-f001:**
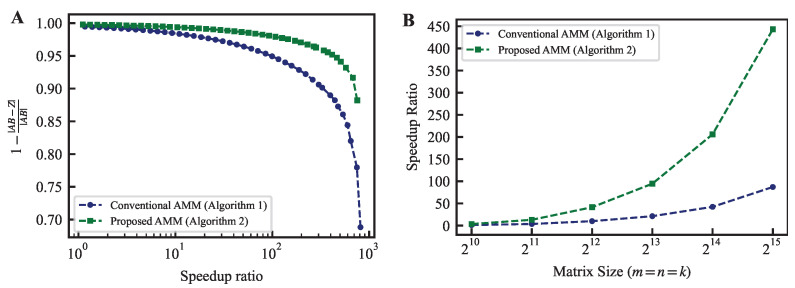
(**A**) The trade-off between the speedup ratio and Frobenius norm error when calculating AMM using the conventional and proposed methods. The matrices are of size $m=n=k=2^{15}$, with elements randomly generated from a uniform distribution between 0 and 1. The speedup ratio is the ratio of the time taken to calculate the matrix product precisely using cuBLAS on a GPU to the time taken to execute AMM. (**B**) Comparison of speedup ratios for achieving a relative accuracy of 0.95. It is clear that as the matrix size increases, the difference in speedup rate increases.

**Figure 2 entropy-25-01130-f002:**
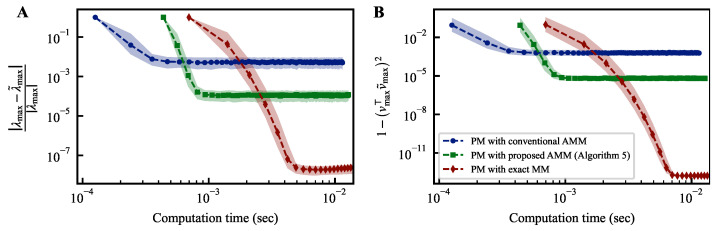
Temporal convergence curves of power methods. The differences between the estimated largest eigenvalue ${\tilde{\lambda }}_{\max}$ and its corresponding eigenvector ${\tilde{v}}_{\max}$ at each iteration of the power method from their respective true values, denoted as $\lambda_{\max}$ and $v_{\max}$, are plotted on the vertical axis of each figure.

**Figure 3 entropy-25-01130-f003:**
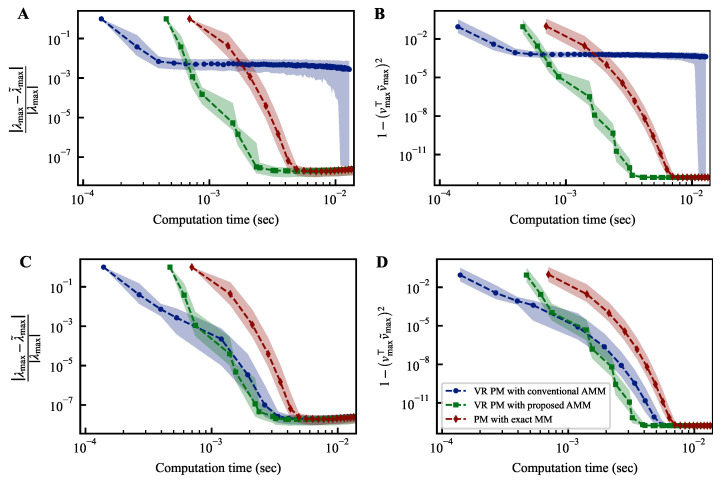
Temporal convergence curves of variance reduction power methods. The differences between the estimated largest eigenvalue ${\tilde{\lambda }}_{\max}$ and its corresponding eigenvector ${\tilde{v}}_{\max}$ at each iteration of the power method from their respective true values, denoted as $\lambda_{\max}$ and $v_{\max}$, are plotted on the vertical axis of each figure. The top row (**A**,**B**) corresponds to $\theta =1-10^{-4}$, while the bottom row (**C**,**D**) corresponds to $\theta =1-10^{-3}$. The notation θ is a parameter representing the threshold for switching between exact and approximate matrix products in power methods; the details of which are described in the main text.

## Data Availability

Readers can download the programs used in Section 3.1 and Section 3.2 from https://github.com/Takuya-Okuyama/AMM_with_preprocessing (accessed on 24 July 2023) and https://github.com/Takuya-Okuyama/eigenvalueComputation_by_AMM (accessed on 24 July 2023), respectively.

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
