# Peer review of "Acceleration of Approximate Matrix Multiplications on GPUs"

_entropy, 2023, doi:10.3390/e25081130_

Round 1

Reviewer 1 Report

In this work, the authors propose an approximate method for matrix multiplication. Approximate computing is a very hot research topic, especially in machine learning and artificial intelligence, where matrix multiplication is one of the primary operations. The article is well-written, and the results are convincing. That being said, I still have some suggestions for improvement.

As the authors note, approximative computing, especially multiplication, is used today on several different layers of computer systems. One of the very appropriate layers for the approximation of multiplication is the hardware layer, where recently, there has been much research in the field of the realization of approximate multipliers and their use in matrix multiplication in the so-called hardware tensor units intended for the GEMM (General Matrix Multiplication) operation. With approximation in hardware, it is primarily possible to reduce the energy consumption of digital circuits used in machine learning (e.g. tensor cores in GPUs). The authors should expand the introductory chapter by discussing this area and what advantages and improvements their approach brings compared to the hardware approximation of multiplication. Is it possible to reduce the power consumption in GPUs with the proposed method? I suggest some current works in this field:

Lee, K.J.; Lee, J.; Choi, S.; Yoo, H.J. The Development of Silicon for AI: Different Design Approaches. IEEE Trans. Circuits Syst. I Regul. Pap. 2020, 67, 4719–4732.

Pilipović, R.; Risojević, V.; Božič, J.; Bulić, P.; Lotrič, U. An Approximate GEMM Unit for Energy-Efficient Object Detection. Sensors 2021, 21, 4195. https://doi.org/10.3390/s21124195

Kim, M.S.; Barrio, A.A.D.; Oliveira, L.T.; Hermida, R.; Bagherzadeh, N. Efficient Mitchell’s Approximate Log Multipliers for Convolutional Neural Networks. IEEE Trans. Comput. 2019, 68, 660–675.

Ansari, M.S.; Cockburn, B.F.; Han, J. An Improved Logarithmic Multiplier for Energy-Efficient Neural Computing. IEEE Trans. Comput. 2021, 70, 614–625

R. Pilipović, P. Bulić and U. Lotrič, "A Two-Stage Operand Trimming Approximate Logarithmic Multiplier," in IEEE Transactions on Circuits and Systems I: Regular Papers, vol. 68, no. 6, pp. 2535-2545, June 2021, doi: 10.1109/TCSI.2021.3069168.

Kim, M.S.; Del Barrio Garcia, A.A.; Kim, H.; Bagherzadeh, N. The Effects of Approximate Multiplication on Convolutional Neural Networks. IEEE Trans. Emerg. Top. Comput. 2021, 1

Author Response

Response to Reviewer 1 Comments

Point 1: As the authors note, approximative computing, especially multiplication, is used today on several different layers of computer systems. One of the very appropriate layers for the approximation of multiplication is the hardware layer, where recently, there has been much research in the field of the realization of approximate multipliers and their use in matrix multiplication in the so-called hardware tensor units intended for the GEMM (General Matrix Multiplication) operation. With approximation in hardware, it is primarily possible to reduce the energy consumption of digital circuits used in machine learning (e.g., tensor cores in GPUs). The authors should expand the introductory chapter by discussing this area and what advantages and improvements their approach brings compared to the hardware approximation of multiplication. Is it possible to reduce the power consumption in GPUs with the proposed method? I suggest some current works in this field:

  • Lee, K.J.; Lee, J.; Choi, S.; Yoo, H.J. The Development of Silicon for AI: Different Design Approaches. IEEE Trans. Circuits Syst. I Regul. Pap. 2020, 67, 4719–4732.
  • Pilipović, R.; Risojević, V.; Božič, J.; Bulić, P.; Lotrič, U. An Approximate GEMM Unit for Energy-Efficient Object Detection. Sensors 2021, 21, 4195. https://doi.org/10.3390/s21124195
  • Kim, M.S.; Barrio, A.A.D.; Oliveira, L.T.; Hermida, R.; Bagherzadeh, N. Efficient Mitchell’sApproximate Log Multipliers for Convolutional Neural Networks. IEEE Trans. Comput. 2019, 68,660–675.
  • Ansari, M.S.; Cockburn, B.F.; Han, J. An Improved Logarithmic Multiplier for Energy-Efficient Neural Computing. IEEE Trans. Comput. 2021, 70, 614–625
  • Pilipović, P. Bulić and U. Lotrič, "A Two-Stage Operand Trimming Approximate LogarithmicMultiplier," in IEEE Transactions on Circuits and Systems I: Regular Papers, vol. 68, no. 6, pp.2535-2545, June 2021, doi: 10.1109/TCSI.2021.3069168.
  • Kim, M.S.; Del Barrio Garcia, A.A.; Kim, H.; Bagherzadeh, N. The Effects of Approximate Multiplication on Convolutional Neural Networks. IEEE Trans. Emerg. Top. Comput. 2021, 1

Response 1: Comment 1 is remarkably supportive and insightful. As reviewer 1 rightly points out, the energy efficiency of approximate computations is indeed of paramount importance. With this in mind, I have added a fourth paragraph on the energy performance improvements in matrix computations achieved by custom hardware. I also explained the potential of our proposed method to reduce energy consumption.

Reviewer 2 Report

The contribution of the paper is twofold: first, it presents an improved version of a Monte Carlo method for approximate matrix multiplication (AMM) by an additional preprocessing step and proves some results on the accuracy of the method; second it experimentally evaluates the the improved method by various benchmarks of a CUDA implementation, in particular with respect to the trade-off between speedup and accuracy in comparison to the base method. It also evaluates the implementation in the context of practical applications, focusing on the problem of computing eigenvalues and eigenvectors in Principal Component Analysis. It is shown that the improved method indeed yields significant benefits. The paper is very well written, presents significant results, and very thoroughly evaluates them. Thus I recommend acceptance.

Minor issues:

* Algorithm 2 presented at the end of Section 2 refers to various notions (alpha, beta, A', B') that are introduced at the very beginning of the section. I recommend to already present this algorithm and the corresponding explanations (last two paragraphs of Section 2) immediately before the theorems, giving here a forward outlook on Theorem 9 and its relevance for the algorithm. By such an organization, the main results can be presented in a compact form on the first two pages of Section 2; this motivates better the remainder of the section with the theorems and their proofs (which can be skipped by the impatient reader, at least on first reading).

* line 202: "determined" -> "determine"

Author Response

Response to Reviewer 2 Comments

Point 1: Algorithm 2 presented at the end of Section 2 refers to various notions (alpha, beta, A', B') that are introduced at the very beginning of the section. I recommend to already present this algorithm and the corresponding explanations (last two paragraphs of Section 2) immediately before the theorems, giving here a forward outlook on Theorem 9 and its relevance for the algorithm. By such an organization, the main results can be presented in a compact form on the first two pages of Section 2; this motivates better the remainder of the section with the theorems and their proofs (which can be skipped by the impatient reader, at least on first reading).

Response 1: Thank you for your valuable feedback. Indeed, it would be more reader-friendly to present the results comprehensively at the beginning of section 2. As a result, I have relocated the pseudocode (Algorithm 2) for the proposed method and its corresponding explanations before Theorem 1. In addition, I have made necessary revisions and expansions to ensure overall consistency throughout the text.

Point 2: line 202: "determined" -> "determine"

Response 2: As you rightly pointed out, I have made the necessary revisions.
